# An Analysis of Stopping Strategies in Conversational Search Systems

## ABSTRACT

Stopping strategies are a crucial aspect of conversational systems and user simulations, as they provide insight into when users end their interactions, which is vital for creating realistic simulations. While the Information Retrieval (IR) community has studied this topic extensively, little research has been done on stopping strategies in Conversational Search Systems (CSSs). This is due to conversations' unique sequential and interactive nature, where traditional IR techniques struggle to accurately predict stopping points well and require new methods to be adapted from traditional IR techniques. In this paper, we adapt Stopping Rules (SRs) from the IR community to the conversational setting, creating new SRs and identifying core features for each. We then analyze these features with several conversational datasets and aim to identify key features that predict stopping points in conversations between users and CSSs. We found that models based on these features performed well in predicting stopping points and that textual statistical features, i.e., numbers of *words*, *nouns*, *noun phrases* and *sentences* users received from systems or outputted by users, always play a significant role in determining stopping points, with the number of outputted *unique nouns* playing a particularly important role as an SR. Our results provide a foundation for developing more realistic user models and simulators and guiding the design of more reliable evaluation measures for CSSs.

## CCS CONCEPTS

• **Information systems** → **Users and interactive retrieval**; Evaluation of retrieval results.

## KEYWORDS

Dialogue Systems, User Modeling, Stopping, Conversational Search

**ACM Reference Format:**
Anonymous Author(s). 2023. An Analysis of Stopping Strategies in Conversational Search Systems. In *Woodstock '23: ACM Symposium on Neural Gaze Detection, June 03–05, 2023, Woodstock, NY*. ACM, New York, NY, USA, 11 pages. https://doi.org/XXXXXXX.XXXXXXX

## 1 INTRODUCTION

In traditional search systems, queries are generally handled as independent entities, with the exception of ad hoc retrieval. This approach means that the response to a particular query is not influenced by the responses to previous queries. However, in Conversational Search Systems (CSSs), queries form sequences like conversations, making this independence assumption unrealistic. This dependency between queries makes the design of user simulators more challenging, as it is important to take into account features such as the recency effect, which refers to what happened in the last few turns near the end of a conversation, as it plays a crucial role in predicting user satisfaction [26]. Without realistic models of user-stopping behaviour, such an effect cannot be simulated, making the modelling of stopping behaviour essential to understanding user behaviour and developing user simulators. If these models don't incorporate realistic stopping points, it becomes challenging to simulate critical aspects like the recency effect, which significantly influences user satisfaction.

In this paper, we focus on the stopping of conversations between users and a CSS. A CSS provides information to users via a conversational interface using natural language [27]. While the field of CSSs is well-established, with connections to Information Retrieval (IR) systems, chatbots, and dialogue systems, the evaluation of CSSs remains an open question [4, 17, 37]. Simulating a conversation is a promising approach for developing new evaluation methods, as it allows for low-cost and repeatable experimentation [5]. However, it is still a challenging task due to the complexity and uncertainty of real user interactions. One of the main challenges is determining the appropriate stopping point, as stopping too early can result in missing information, while stopping too late can lead to redundant or misleading information. As noted in Section 2 based on previous studies, the decision to stop often reflects the user's sentiment. In the IR community, several *stopping Rrules* (SRs) (or *Stopping Strategies*) have been proposed to depict stopping behaviours. SRs model users stopping behaviours according to given conditions. For example, one of the most classic SRs is Fixed Depth, which assumes users will stop their conversations once they are longer than a fixed length. These studies aim to quantify the feeling of "enough". For example, users would likely stop the conversation when they are frustrated as well as they are satisfied.

Modelling and simulating stopping behaviour is of great importance in the field of CSS, and understanding it is crucial for evaluating search behaviour and performance [39, 40]. In this paper, we analyze SRs and features for each SR using four datasets detailed in Section 5: TopiOCQA, FAITHDIAL, TREC CAsT 2021 and EvalCran. For features based on the augmented four datasets, we run logistic regression to delve into weights and features for further explanation. The goal of this paper is to study SRs and their features in a conversational setting and predict the users' stopping points. We then analyze the weights of features on different datasets. We also identify interesting features by combining them separately and evaluate the performance of models based on these features. Our contributions can be summarized as follows:

- An analysis of the performance of SRs on various datasets in a conversational setting.
- An analysis of the features that contribute to predicting stopping points.
- Identification of two distinct approaches to modeling stopping behaviors in conversations.

Our results provide a foundation for developing more realistic user models and simulators, as well as guiding the design of more reliable evaluation measures for CSSs.

## 2 RELATED WORK

### 2.1 Conversational Search

Conversational search is increasing in popularity due to the usefulness of searching on modern devices with small or no screens [27]. As defined by Anand et al. [4], CSSs inherit properties from IR systems, chatbots and dialogue systems, such as access to information, statefulness, and interaction naturalness. Therefore, many studies in this field focus on user experience [3, 7] and performance [4, 24, 28, 49, 51]. Additionally, many studies also focus on the usage of information [2, 11].

Despite the many studies in this field, the evaluation of CSSs is still relatively underdeveloped [12, 30, 37]. This remains a key challenge for CSSs [4]. Traditional metrics, such as nDCG and MRR, and metrics from other fields, such as BLEU and ROUGE, are still popular in evaluating CSS in some studies [18, 30, 31, 43, 46]. However, recent studies have shown that these metrics may not accurately reflect the real user experience and preference [32, 38].

This study aims to contribute to the IR community by improving stopping prediction, which is essential for accurately simulating user behaviour.

### 2.2 Modelling and Simulating Users' Behaviour

In IR, two popular methods for simulating users are cognitive and statistical approaches [23]. Cognitive approaches, which were first introduced by Belkin [6], focus on characterizing users by their objectives, problems, and knowledge of the world. This approach has been further developed by many studies (see [22, 29, 35]). On the other hand, statistical approaches analyze user behaviours and satisfaction [9, 13, 16, 19, 23]. These user predictive models are also commonly used as the basis for simulating users in IR.

Early simulators in the field of conversational systems relied on statistical models [21, 48]. These simulators were built around a fixed corpus, and thus, had limitations when it came to the diversity of user intents [23]. To overcome these limitations, agenda-based simulation became popular, as it allowed for more realistic responses and the ability to easily construct dialogue strategies [33, 36, 47]. Recently, there has been a trend towards using deep learning models for simulating users, such as training models with adversarial generative approaches [52], reinforcement learning approaches [10], and extracting abstract knowledge from data using inverse reinforcement learning [8]. Ideas from psychology communities, such as Priming Effects, are also considered as pathways to simulate user actions [25].

In this study, the primary focus is on the stopping of conversation, a crucial aspect that has not been adapted to recent trends. Various

SRs will be analyzed and modified to enhance the prediction of conversation termination.

### 2.3 Stopping Rules

Determining when to stop a conversation is a complex task as it is influenced by various features, including the user's cognitive state and decision-making processes [39]. Despite the difficulties of modelling stopping behaviours, several SRs have been proposed in the IR community to explain when users stop. Cooper [15] presented two rules:

- the frustration point rule, which states that users will stop after receiving a certain number of non-relevant replies; and
- the satisfaction stopping rule, which states that users will stop only after receiving a certain number of relevant replies.

Kraft and Lee [34] introduced two additional rules:

- the expected search length rule, which is based on the number of results read by users; and
- the combination rule, which combines the rules of frustration and satisfaction, stating that users will stop if they are satisfied or disgusted by receiving too many non-relevant replies.

Nickles [42] then proposed four cognitive rules:

- the mental list rule, which assumes that users have a list of criteria that must be satisfied before stopping;
- the representational stability rule, where a user continues to query until the underlying mental model begins to stabilize;
- the difference threshold rule, a user sets an a priori difference level to gauge when there are no new replies, and;
- the magnitude threshold rule, a user has to attain enough information before stopping.

In this study, we aim to develop SRs that build upon those established in previous studies. In addition to the pre-existing SRs, we are introducing a range of indicators derived from lexical context. These indicators are significant as they have the potential to mirror user preferences within a conversation. [50]

## 3 RESEARCH QUESTIONS

In this paper, we aim to answer the following research questions:

**RQ1.** *How do different SRs perform on different datasets?*

To answer this question, we will evaluate the performance of various SRs using appropriate measures. This is crucial for understanding the reliability of these rules and the features associated with them in predicting stopping points.

**RQ2.** *What is the performance of models built on selected features?*

To build and evaluate models, we will select relevant features from SRes and augment datasets from completed conversations. We will also test the performance of models trained on one dataset on other datasets to demonstrate the robustness of these features and models.

**RQ3.** *How do the features influence the prediction of stopping?*

By analyzing different features and datasets, we will investigate if there are any unique or particularly important features that contribute to the stopping behaviours. This will provide insights into other aspects of stopping that can be further explored.

## 4 STOPPING RULES

In this section, we propose several SRs. The first SR, which we also consider as a baseline, is the Fixed Depth.

**SR1 Fixed Depth** Users will stop once the conversation is longer than a given length $l$, where $l$ is a feature.

This SR is a basic approach in many simulations and its performance depends on the specific scenario [40]. For example, this SR will perform well in simulating conversations with a predefined number of turns. However, it may not be suitable for other settings.

We also propose **SR2-SR7**, which are based on rules of frustration and satisfaction from previous studies.

**SR2 Similar Hits** Users will stop once they get a response similar to a previous response. The feature of this SR is the number of similar (more than 0.5, TF-IDF) responses $s$.

**SR3 Continually Similar** Users will stop once they get a series of similar responses. The feature of this SR is the number of series of similar (more than 0.5, TF-IDF) responses $s_c$.

**SR2** and **SR3** focus on similar system's responses, which are based upon the difference threshold rule [42]. For these two SRs, the similarity between the current response and the previous response is calculated via TF-IDF. Getting similar responses means getting less new information, thereby frustrating the users and resulting in stopping.

**SR4-SR7** take another approach to judge frustration and satisfaction, which is based on the relevance of the responses. We introduce these SRs because the relevance of responses from the CSS has been proved that plays a significant role in satisfaction [26]. However, these SR and factors will only be applied in datasets where relevance labels are available.

**SR4 Total Non-relevant** Users will stop once they receive more than a number of non-relevant responses. The feature of this SR is the number of non-relevant responses $i$.

**SR5 Continually Non-relevant** Users will stop once they observe a series of non-relevant responses. The feature of this SR is the number of such series $ii$.

**SR6 Total Relevant** Users will stop once they receive more than a number of relevant responses. The feature of this SR is the number of relevant responses $r$.

**SR7 Relevant-rate** Users will stop once the relevant rate of replies is lower than a given number. The feature of this SR is the relevant-rate $P(rel)$.

In **SR4** and **SR5**, we focus on the number of non-relevant responses and assume that frustration accumulates with each non-relevant response received, leading to a stopping point [15]. Similarly, in **SR6**, we posit that users will feel satisfied when they receive a sufficient number of relevant responses. Lastly, with **SR7**, we consider the overall quality of the conversation, assuming that users will stop if they believe the quality is lower than their expectation.

We then design **SR8-SR10** to analyze conversations from a statistical perspective. These SRs are based on statistics calculated from the conversation logs. We assume that the number of words used by the replies reflects the effort users put into the conversation. We assume the same for the number of sentences, which may reflect the number of information clusters in the conversation.

**SR8 Total Received** Users will stop once they have received enough information:

**SR8.1 Words** Users will stop after reading a number of words. The feature of this SR is the number of received words $n_{wtr}$.

**SR8.2 Nouns** Similar to **SR8.1** but only considering nouns. The feature of this SR is the number of received nouns $n_{ntr}$.

**SR8.3 Sentences** Similar to **SR8.1** but counting sentences. The feature of this SR is the number of received sentences $n_{str}$.

**SR9 Total Sent** Users will stop after replying with enough information:

**SR9.1 Words** User will stop once they replied with a number of words. The feature of this SR is the number of words sent $n_{wts}$.

**SR9.2 Nouns** Similar to **SR9.1**, but counting the number of nouns. The feature of this SR is the number of nouns sent $n_{nts}$.

**SR9.3 Sentences** Similar to **SR9.1**, but counting the number of sentences. The feature of this SR is the number of sentences sent $n_{sts}$.

**SR10 Last Received** Users will stop once they have received enough information in the last turn:

**SR10.1 Words** User will stop once they receive a number of words from the system in the last turn. The feature of this SR is the number of received words in the last turn $n_{wlt}$.

**SR10.2 Nouns** Similar to **SR10.1**, but counting the number of nouns. The feature of this SR is the number of received nouns in the last turn $n_{nlt}$.

**SR10.3 Sentences** Similar to **SR10.1**, but counting the number of sentences. The feature of this SR is the number of received sentences in the last turn $n_{slt}$.

Note that when considering nouns (SR8.2, SR9.2 and SR10.2), we count the number of unique nouns/noun phrases as a measure of the amount of new information conveyed in the conversation as Clark and Sengul [14] point out the strong relationship between understanding new information and identifying new nouns/noun phrases. We used TextBlob and NLTK packages to locate nouns/noun phrases in this paper.

In this study, we will utilize all of the SRs defined in this section to identify stopping points. Initially, each rule will be analyzed individually. Subsequently, indicators derived from these SRs will be employed as features in the development of models aimed at predicting stopping points.

## 5 DATA

In this study, we analyze four datasets to understand the stopping behaviours of users in conversational search systems. We choose four basic datasets in order to introduce more settings. For example, we use two datasets that include replies in free-form (TopiOCQA and FAITHDIAL), and two datasets that include replies with paragraphs. On average, the FAITHDIAL dataset has the shortest conversation depth, with an average of 4.5 turns, while the TopiOCQA dataset has the longest conversation depth, with an average of 13

**Table 1: Properties of each dataset. Size refers to the total number of conversations, $\bar{l}$ refers to the average depth of the dataset, $\sigma$ refers to the standard deviation on depth, and Rel refers to if the relevance of responses is labelled.**

| Dataset | Size | $\bar{l}$ | $\sigma$ | Answer Style | Rel |
|---------|------|-----------|----------|--------------|-----|
| TopiOCQA | 3920 | 13 | 3.3 | Free-form | F |
| FAITHDIAL | 5649 | 4.5 | 0.5 | Free-form | F |
| TREC CAsT 2021 | 26 | 9.2 | 1.7 | Passage | T |
| EvalCran | 131 | 5.4 | 2.6 | Passage | T |
| Topi+FD | 9569 | 7.8 | 4.6 | Free-form | F |
| TREC+Cran | 157 | 6.7 | 2.7 | Passage | T |
| ALL | 9726 | 7.8 | 4.6 | Mixed | F |

turns. Replies from FAITHDIAL and TopiOCQA are written by the system as free-form answers.

## 5.1 Datasets

***TopiOCQA (Topi).*** TopiOCQA is a conversational dataset that draws its content from Wikipedia [1] and is characterized by its focus on the complexities of topic switching. Each conversation turn includes a marker indicating the topic being discussed. Comprising of 3,920 interactions, the dataset includes free-form answers. Most of them are within one sentence. These conversations have a conversational depth of 13 turns on average, with a standard deviation of 3.3 and four distinct topics.

***FAITHDIAL (FD).*** FAITHDIAL is a conversational dataset based on Wikipedia [20] that is designed to study the phenomenon of information seekers and bots misunderstanding each other. It includes 5,649 conversations, comprising 50,761 queries and responses. The bot typically responds to the user with a few sentences per turn. The average length of conversations in this dataset is 4.5, with a standard deviation of 0.5.

***TREC CAsT 2021 (TREC).*** The TREC Conversational Assistant Track (CAsT) 2021 dataset [1] offers a collection of open-domain, information-centric conversational dialogues featuring 26 conversations between users and systems. TREC CAsT 2021 emphasizes the importance of considering the context of the dialogue and retrieving relevant information. The dataset includes information from sources such as the Washington Post[2] (2012-2020), KILT Wikipedia (August 1st, 2019) [44], and MS MACRO [41]. The system provides a passage in response to the user's question in each turn. The average conversation length in this dataset is 9.2, with a standard deviation of 1.7.

***EvalCran (Cran).*** This dataset, collected by Lipani et al. [37], consists of 131 conversations based on the SQuAD dataset [45]. It features a user-system interaction where the user poses a query to the system, and the system responds with a paragraph. Users then evaluate the relevance of the paragraph to the query.

---
[1]https://www.treccast.ai/
[2]https://trec.nist.gov/data/wapost/

## 5.2 Grouping, Splitting and Augmenting

In Table 1, we show the properties of each dataset. In order to make the most of the features present in each dataset, we group them to create new datasets. For example, we merge TopiOCQA and FAITHDIAL to augment diversity and also merge EvalCran and TREC CAsT 2021. We merged TopiOCQA, and FAITHDIAL since both of them feature free-form as the response style, as well as TREC CAsT 2021 and EvalCran feature raw passages as the response. Another reason is the size of datasets, where TopiOCQA is similar to FAITHDIAL while TREC is similar to EvalCran. Additionally, we combine all four datasets into one to take advantage of all available data. In Table 1, we also show the statistics of the merged datasets.

To train the Logistic Regression (LR) models, we augmented the original datasets by producing unfinished conversations, i.e., negative samples. This is done by randomly cutting the conversations at different points. For each original conversation, there is one corresponding unfinished version for a balanced distribution. In the final dataset, 50% of the conversations are unfinished. Finally, we divide each dataset randomly into 7:3 train and test sets.

The process of generating datasets in this study is 1) Merging datasets (e.g., TREC CAsT 2021 and EvalCran) into one dataset if needed; 2) Shuffling the dataset; 3) The dataset is split into a train set and a test set with a 7:3 ratio, and; 4) Augmenting each set by generating unfinished versions.

To summarize, we have four original datasets: TopiOCQA, FAITHDIAL, EvalCran, and TREC CAsT 2021. By grouping them, we create three mixed datasets: Topi+FD, TREC+Cran and ALL. Each dataset is then further split into train and test sets and augmented by unfinished conversations.

## 6 EXPERIMENTS AND FINDINGS

In this study, we approach the task of understanding user behaviour in conversations as a binary classification problem. At each turn, we aim to predict whether the user will continue or stop the conversation. To make this prediction, we utilize as features the previously defined SRs. We employ Logistic Regression (LR), a method well-suited for binary classification tasks, due to its efficiency and ability to illustrate the impact of individual features.

To answer the research question **RQ1**, which asks about the performance of different SRs on different datasets, we study the performance of each SR by taking each one of them individually and use it to make predictions about whether users stop or not.

To address **RQ2** about the predictability of users' stopping behaviour combining all features and **RQ3** about understanding the importance of each feature, we instead combine all the parameters and train an LR model and utilize the LR weights to analyse the impact of each SR on the prediction. The greater the weight, the greater the influence of the stopping rule. Positive weights indicate a tendency towards stopping, while negative weights suggest a tendency towards continuing.

To evaluate the performance of the trained models, we use the metric Precision, Recall, F1-score and AUC (Area Under the Precision-Recall Curve). We use the F1-score as the main measure, and the results for the other measures are attached in the Appendix.

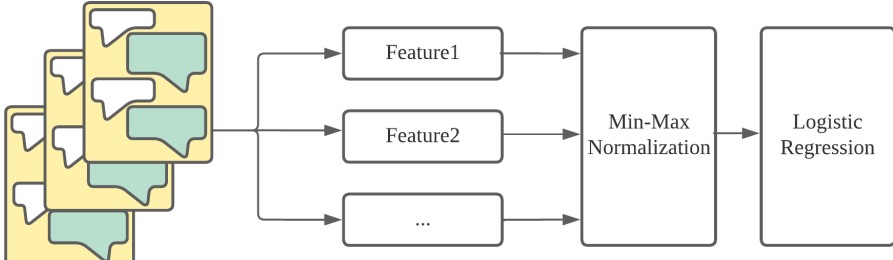

**Figure 1: From the conversations, features are extracted. Then, the features of the conversations used for the training set are normalized using a Min-Max normalization and used to train Logistic Regression models.**

**Table 2: Mapping between SR and Features.**

| SR | Feature | Symbol |
|---|---|---|
| SR1 | Number of depth | $l$ |
| SR2 | Number of similar replies | $s$ |
| SR3 | Number of clusters of similar replies | $s_c$ |
| SR4 | Number of received non-relevant replies | $i$ |
| SR5 | Number of continually non-relevant lists | $ii$ |
| SR6 | Number of received relevant replies | $r$ |
| SR7 | Ratio of relevant replies and total replies | $P(rel)$ |
| SR8.1 | Number of total received words | $n_{wtr}$ |
| SR8.2 | Number of total received unique nouns | $n_{ntr}$ |
| SR8.3 | Number of total received sentences | $n_{str}$ |
| SR9.1 | Number of total sent words | $n_{wts}$ |
| SR9.2 | Number of total sent unique nouns | $n_{nts}$ |
| SR9.3 | Number of total sent sentences | $n_{sts}$ |
| SR10.1 | Number of received words in the last turn | $n_{wlt}$ |
| SR10.2 | Number of received unique nouns in the last turn | $n_{nlt}$ |
| SR10.3 | Number of received sentences in the last turn | $n_{slt}$ |

## 6.1 LR Model Training

In Figure 1, we show the process of training LR models. From the raw conversations, we calculate the features. After a Min-Max normalization, these features are used to train the LR models. As concluded in Section 5, the datasets will be augmented with generated unfinished conversations and keep 50% of conversations unfinished. For each conversation, each feature will be calculated.

Because TREC CAsT 2021 and EvalCran are the only datasets with relevant labels on systems' replies, we trained two LR models for them. One with features related to relevance and one without.

## 6.2 Performance of SRs

To answer **RQ1**, we use single features from SRs to train LR models. Table 2 shows the features of each SR we adjust to adapt different datasets.

The F1-scores of each model on the corresponding test set are shown in Table 3. The results indicate that the performance of SR varies depending on the dataset used. For instance, using the number of total sentences sent by users can yield high performance

on TREC and Topi datasets, while on other datasets, the F1-score is lower. This highlights the limitations of traditional SRs, as the most classic rule, the fixed depth $l$, only achieves good performance on two datasets. The rules based on similarity are only effective on TREC and Cran, which may be due to the fact that replies on Topi and FD are generated by conversational systems rather than other humans. The rules based on relevance are also limited by the dataset, where only Cran and TREC have users' replies labelled. Overall, only one SR achieved good performance on each dataset. The feature $n_{nts}$, the number of nouns sent by users, achieved scores higher than 0.7 on all datasets, with the exception of a score of 0.645 on Topi.

To address **RQ2**, we trained LR models using all features on all seven datasets discussed in Section 5. We then applied all of the trained models on each dataset to evaluate their capacity and identify any biases resulting from training on a single dataset. After training the models, we analyzed the weights of each feature to gain insight into the models. We will begin by examining the weights compared with other features in all eight models to establish a general ranking of features for each dataset.

In Table 4, we display the scores of the LR models on the test sets. For the EvalCran and TREC CaST 2021 datasets, two models were trained: one without relevance features (Cran, TREC) and one with relevance labels (Cran+, TREC+). The first observation is that all models achieved high scores on the corresponding test sets, indicating that combining all the features listed in Section 4 allows the models to effectively extract relevant information from the context to predict stopping points. Secondly, the scores of Cran are lower than those of Cran+ as well as TREC vs TREC+, suggesting that features related to relevance play an important role in predicting stopping points and cannot be replaced by other features listed in Section 4.

In Table 5, we present the F1-scores of models that were trained and tested on different datasets. The columns indicate the source of the test sets, while the rows indicate the source of the training sets. Here we discover that four models obtained satisfactory F1-scores (higher than 0.7) in the majority of the test sets. This suggests that the weight distributions of these four models (All, Topi+FD, FD, and Cran) effectively predict stopping points in diverse settings.

**Table 3: The F1 scores of LR models. In bold we indicate all features achieving a score equal to or greater than** 0.700**.**

| Feature | ALL | | Topi+FD | | Topi | | FD | | TREC+Cran | | TREC | | EvalCran | |
|---|---|---|---|---|---|---|---|---|---|---|---|---|---|---|
| | | | | Free-Form | | | | | | | Passage | | | |
| $l$ | 0.489 | (7) | 0.487 | (8) | **0.789** | (3) | 0.000 | (11) | 0.583 | (6) | **0.800** | (3) | 0.526 | (5) |
| $s$ | 0.271 | (10) | 0.259 | (11) | 0.512 | (8) | 0.007 | (10) | 0.667 | (4) | 0.400 | (9) | **0.714** | (3) |
| $s_c$ | 0.131 | (11) | 0.108 | (12) | 0.237 | (11) | 0.007 | (10) | **0.757** | (1) | **0.857** | (2) | **0.733** | (1) |
| $n_{wtr}$ | 0.681 | (4) | 0.682 | (5) | **0.752** | (5) | 0.616 | (4) | 0.653 | (5) | 0.667 | (6) | 0.650 | (4) |
| $n_{ntr}$ | 0.515 | (6) | 0.510 | (7) | **0.778** | (4) | 0.125 | (9) | 0.653 | (5) | 0.667 | (6) | 0.650 | (4) |
| $n_{str}$ | **0.748** | (2) | **0.751** | (2) | 0.692 | (6) | **0.790** | (1) | 0.653 | (5) | 0.667 | (6) | 0.650 | (4) |
| $n_{wts}$ | **0.753** | (1) | **0.759** | (1) | **0.817** | (2) | **0.709** | (3) | 0.211 | (10) | 0.667 | (6) | 0.000 | (9) |
| $n_{nts}$ | **0.711** | (3) | **0.710** | (3) | 0.645 | (7) | **0.748** | (2) | **0.743** | (2) | **0.857** | (2) | **0.714** | (3) |
| $n_{sts}$ | 0.681 | (4) | 0.685 | (4) | **0.830** | (1) | 0.533 | (6) | 0.316 | (6) | **1.000** | (1) | 0.000 | (9) |
| $n_{wlt}$ | 0.478 | (8) | 0.473 | (9) | 0.355 | (10) | 0.526 | (7) | 0.653 | (5) | 0.667 | (6) | 0.650 | (4) |
| $n_{nlt}$ | 0.393 | (9) | 0.385 | (10) | 0.397 | (9) | 0.376 | (8) | 0.667 | (4) | **0.750** | (4) | 0.650 | (4) |
| $n_{slt}$ | 0.605 | (5) | 0.609 | (6) | 0.645 | (7) | 0.578 | (5) | 0.000 | (11) | 0.000 | (10) | 0.000 | (9) |
| $i$ | - | (-) | - | (-) | - | (-) | - | (-) | 0.507 | (7) | **0.714** | (5) | 0.415 | (6) |
| $r$ | - | (-) | - | (-) | - | (-) | - | (-) | 0.698 | (3) | 0.588 | (7) | **0.722** | (2) |
| $ii$ | - | (-) | - | (-) | - | (-) | - | (-) | 0.312 | (9) | 0.500 | (8) | 0.143 | (8) |
| $P(rel)$ | - | (-) | - | (-) | - | (-) | - | (-) | 0.488 | (8) | 0.500 | (8) | 0.413 | (7) |

**Table 4: Scores of LR models on test sets.**

| | ALL | TF | Topi | FD | TC | TREC | Cran | TREC$^+$ | Cran$^+$ |
|---|---|---|---|---|---|---|---|---|---|
| | | Free-Form | | | Passage | | | | |
| $F1$ | 0.806 | 0.815 | 0.857 | 0.944 | 0.809 | 0.706 | 0.694 | 0.973 | 0.769 |
| $Precision$ | 0.797 | 0.812 | 0.812 | 0.894 | 0.776 | 0.667 | 0.714 | 0.947 | 0.789 |
| $Recall$ | 0.814 | 0.819 | 0.907 | 1.000 | 0.844 | 0.750 | 0.676 | 1.000 | 0.750 |
| $AUC$ | 0.846 | 0.851 | 0.856 | 0.969 | 0.847 | 0.842 | 0.706 | 0.969 | 0.744 |

**Table 5: F1 scores of models on datasets. Columns represent where the test set comes from, while rows represent the origin of the train set.**

| train \ test | ALL | Topi+FD | Topi | FD | TREC+Cran | TREC | Cran |
|---|---|---|---|---|---|---|---|
| | | Free-Form | | | Passage | | |
| ALL | **0.797** | **0.800** | **0.829** | **0.779** | 0.692 | **0.765** | 0.650 |
| Topi+FD | **0.812** | **0.815** | **0.818** | **0.812** | **0.728** | **0.743** | **0.717** |
| Topi | 0.525 | 0.524 | **0.857** | 0.122 | 0.542 | **0.825** | 0.436 |
| FD | **0.846** | **0.849** | **0.737** | **0.944** | **0.769** | **0.743** | **0.784** |
| TREC+Cran | 0.504 | 0.498 | **0.837** | 0.026 | **0.809** | **0.847** | **0.724** |
| TREC | 0.290 | 0.290 | 0.583 | 0.001 | 0.307 | **0.706** | 0.101 |
| EvalCran | **0.738** | **0.738** | **0.825** | 0.656 | **0.742** | **0.800** | 0.694 |

In Table 6, we show the weights of models in Table 4. For each column, the top 5 biggest weights in absolute value are marked in bold, and their orders are also labelled. One finding is that there are at least two distribution patterns of weights in the four high-diverse models. First is ALL and Topi+FD, where features related to the total received from the system and the total outputted by users play a leading role. Second is FD and Cran, where depth is the most significant contributor. Especially in FD, the weight of depth is far larger than the other features. While in ALL and Topi+FD,

the depth is a minor feature. Another finding is that in all four original datasets' models, depth always plays the most important role. This means that the high value of the depth to predict stopping points works well only in specific contexts or that these datasets lack variety in the sampled conversations.

## 6.3 Special Features

Based on previous experiments, we can answer **RQ3** by analyzing features with unique behaviours. One interesting property of such

**Table 6: Weights of LR model per dataset, where the top 5 features for each dataset are marked in bold.**

| | ALL | TF | TC | Topi | FD | TREC | TREC⁺ | EvalCran | EvalCran⁺ |
|---|---|---|---|---|---|---|---|---|---|
| $l$ | 0.819 (11) | 1.593 (7) | **1.882** (1) | **4.841** (1) | **15.992** (1) | **1.015** (1) | **0.943** (1) | **1.731** (1) | **1.737** (1) |
| $s$ | -0.910 (10) | 0.172 (11) | **0.871** (5) | 0.318 (6) | -0.495 (7) | 0.160 (11) | 0.195 (12) | 0.291 (9) | **0.815** (5) |
| $s_c$ | -0.058 (12) | 0.107 (12) | 0.427 (8) | 0.070 (11) | 0.358 (9) | 0.458 (6) | -0.134 (15) | 0.568 (7) | 0.284 (13) |
| $n_{wtr}$ | **4.350** (4) | **7.004** (2) | 0.645 (7) | **0.913** (4) | **1.350** (2) | **0.804** (4) | **0.827** (3) | **0.856** (3) | **1.069** (3) |
| $n_{ntr}$ | **-5.819** (3) | **-8.385** (1) | 0.901 (4) | -0.444 (5) | 0.495 (7) | **0.812** (3) | 0.617 (9) | **0.767** (4) | 0.731 (7) |
| $n_{str}$ | **8.860** (1) | **4.410** (4) | 0.666 (6) | 0.034 (12) | **1.012** (3) | **0.487** (5) | 0.655 (7) | **0.732** (5) | 0.664 (8) |
| $n_{wts}$ | **5.924** (2) | **5.106** (3) | -0.153 (10) | **1.775** (3) | 0.525 (6) | 0.363 (9) | 0.639 (8) | **1.070** (2) | **0.951** (4) |
| $n_{nts}$ | **3.907** (5) | **3.401** (5) | **1.317** (2) | -0.140 (10) | **0.659** (4) | 0.351 (10) | 0.480 (10) | 0.722 (6) | 0.479 (9) |
| $n_{sts}$ | 2.045 (8) | 2.467 (6) | **-1.016** (3) | **4.366** (2) | 0.212 (10) | **0.822** (2) | **0.937** (2) | 0.000 (12) | 0.000 (16) |
| $n_{wlt}$ | 2.206 (6) | 0.954 (9) | -0.384 (9) | 0.199 (9) | 0.017 (11) | -0.399 (7) | -0.320 (11) | 0.516 (8) | 0.219 (14) |
| $n_{nlt}$ | -2.184 (7) | -0.439 (10) | -0.082 (11) | -0.223 (8) | -0.378 (8) | -0.371 (8) | -0.085 (16) | 0.284 (10) | 0.407 (11) |
| $n_{slt}$ | -1.530 (9) | -1.247 (8) | -0.141 (12) | 0.257 (7) | **-0.613** (5) | -0.150 (12) | 0.167 (13) | -0.061 (11) | -0.778 (6) |
| $r$ | - (-) | - (-) | - (-) | - (-) | - (-) | - (-) | 0.663 (6) | - (-) | 0.300 (12) |
| $i$ | - (-) | - (-) | - (-) | - (-) | - (-) | - (-) | **0.667** (5) | - (-) | **1.341** (2) |
| $P(rel)$ | - (-) | - (-) | - (-) | - (-) | - (-) | - (-) | -0.138 (14) | - (-) | -0.048 (15) |
| $ii$ | - (-) | - (-) | - (-) | - (-) | - (-) | - (-) | **0.787** (4) | - (-) | -0.432 (10) |
| bias | -3.334 | -3.639 | -1.438 | -3.900 | -11.951 | -2.093 | -2.659 | -2.237 | -1.901 |

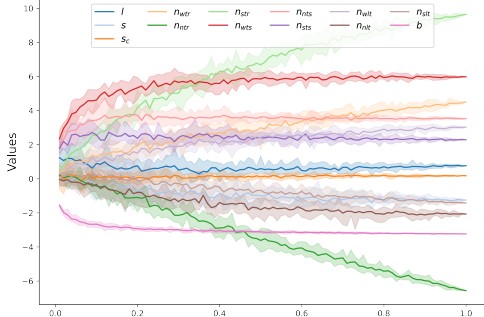

**Figure 2: Weights of features (and bias) in the function of dataset proportion based on 10 runs.**

**Table 7: Test set scores of models using a different proportion of the dataset over 10 runs.**

| Proportion | F1 | Precision | Recall | AUC |
|---|---|---|---|---|
| 10% | 0.798 | 0.802 | 0.794 | 0.840 |
| 20% | 0.796 | 0.810 | 0.782 | 0.843 |
| 30% | 0.794 | 0.806 | 0.783 | 0.845 |
| 40% | 0.793 | 0.809 | 0.778 | 0.846 |
| 50% | 0.797 | 0.811 | 0.784 | 0.845 |
| 60% | 0.800 | 0.809 | 0.790 | 0.846 |
| 70% | 0.797 | 0.809 | 0.785 | 0.847 |
| 80% | 0.799 | 0.810 | 0.788 | 0.847 |
| 90% | 0.801 | 0.810 | 0.791 | 0.848 |
| 100% | 0.801 | 0.811 | 0.791 | 0.848 |

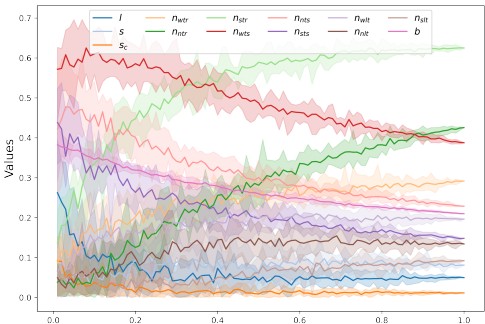

**Figure 3: L2-normalized weights and bias on 10 runs.**

features is when they assume a negative weight, i.e., they promote the conversation to continue. Another potential direction is the changes in features' weights through the change of the dataset size. This is because when the dataset is small, some features may be important just because the LR models are overfitting on them. For example, predicting the stopping points in FD is relatively easy since most of the original conversations stop at turn 4 or 5. Hence depth will play a paramount role. By studying how feature importance changes when increasing the dataset size, which is proportional to the dataset diversity, we show which features tend to become more prominent. Our hypothesis is that when more data is added, features with better generalizability will rise.

Before looking at what happens to the feature weights while changing the proportion of data used, we first look at how the performance of the LR models changes. In Table 7, we show that there is not a significant change in the scores. Therefore, all models trained with a different proportion of data are comparable.

In Figure 2, we show how the features' weights change when changing the proportion of data used. We found that only three features change significantly when the size of the dataset increases,

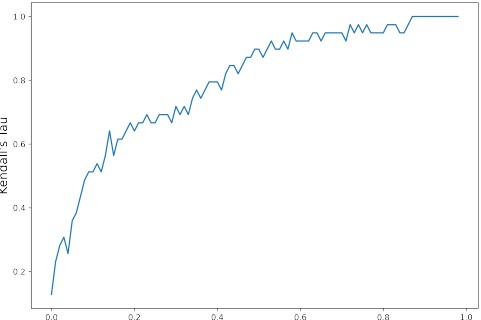

**Figure 4: Kendall's Tau of weights between each step and the final step of Figure 3.**

while other features do not change even after 70% of all data is used. We observe that the features that promote the conversation to continue are: the total number of received unique nouns, the number of unique nouns received at the last turn, the number of sentences received at the last turn and the number of similar replies. However, we notice that the magnitude of the features' weights increases with the increase of the data. For a LR model, this means that the separation of the space between stopping and continuing the conversation is becoming sharper and sharper. This means that without an appropriate normalization, we cannot tell if the features are becoming more important with the dataset size or because their overall magnitude is changing.

To avoid this effect, we also look at how the features' weights change after having them normalized. To do that, we first take the absolute value of each feature, then divide them by the L2-norm of the weights and bias vector. The result of this normalization is shown in Figure 3. After performing this normalization, we observe that the features that become more important are the total number of received sentences, words and unique nouns. The models tend to focus on three specific features when there is a larger sample of data. One notable finding is that the weight of the total number of unique nouns received has a negative and significant increase. Along with the increase in the number of total received sentences, we can deduce that users' stopping behaviour is depicted as a process of receiving information, with each sentence being treated as a unit of information. As users receive more information, they become more inclined to stop. Unique nouns represent novel information, and a larger amount of novel information prolongs the search process and ultimately delays the stopping point.

In Figure 4, we show the Kendall's Tau correlation coefficient of the weights between each dataset proportion vs the full dataset, as shown in Figure 3. This shows that to achieve a Kendall's Tau of 0.85 we need around 50% of the dataset. That indicates that to get a similar distribution of weights as the full dataset, we need around 4,000 conversations.

## 7 DISCUSSION AND CONCLUSION

In this paper, we analyzed several SR and corresponding features. We first adapted SR to fit the conversational setting. Then we analyzed the performance of SR on four independent datasets. After

SR, we further explored features from each SR. We run logistic regressions on different combinations of 4 datasets. Each model from this step is also tested in other combinations of datasets. We then analyzed the weights of features in each model and ran further LR based on different features. Finally, we revealed the special features by changing the amount of input data.

Our results show:

(1) While the baseline (depth, $l$) can achieve good performance, there are more suitable SRs for high-diverse scenarios.
(2) Models with the combination of listed features in Section 4 can properly predict stopping points in different scenarios.
(3) Relevance-related features can contribute to predicting stopping points.
(4) Significant contribution of the depth to predicting stopping points in single scenarios.
(5) The necessarily contribution of features groups of **SR8** and **SR9**.
(6) Two predicting strategies can handle high-diverse datasets: one focuses on features based on text statistics, and another focuses on depth.
(7) Special features where weights will change through increasing diversity, which may also represent some properties of received information.

Result 1 addresses **RQ1**, Result 2 addresses **RQ2**, and Results 3-7 collectively address **RQ3**.

Our research reveals two distinct ways of depicting stopping behaviours. The first approach views stopping as the conclusion of an information-receiving process, where users reach their information limit and feel "enough." Fresh information prolongs this limit, and textual statistical features such as the number of total sentences and unique nouns in received documents provide a reliable representation. This depiction is particularly effective in high-diverse settings. The second approach views stopping as a fixed-length, anticipated journey, where the depth of the conversation is the most important factor. This approach is suitable for single-regular settings.

Our study presents substantial promise in augmenting user simulation and related functions. Specifically, this research can be assimilated into dialogue management systems, offering improved guidance of interactions. Furthermore, the implementation of our findings in agents may facilitate the extension of conversations through more adept selection of ensuing responses. Moreover, simulated users that incorporate our research can achieve more appropriate termination of conversations.

In this study, we identified four limitations from which future works may benefit. First, more SR and features should be considered. For example, the time spent in the conversation is a potential feature for analyzing stopping points. However, in this paper, due to datasets' limitations, these features can not be explored. Second, the datasets are unbalanced. For example, TopiOCQA has 3,920 conversations, while EvalCran has only 131. This may result in a deficient analysis of SR and features in settings related to such datasets. Third, TF-IDF is an effective algorithm in calculating similarities between documents, but in some datasets, the system did not reply with original documents. For example, in FAITHDIAL, candidate documents will be rewritten. In this case, the results of TF-IDF may not be reliable. Finally, Large Language Models like

Llama-2, GPT-3.5, and GPT-4, which are popular within the IR community, can also be utilized for predicting stopping. However, this application is not explored in this study.

## A  OTHER MEASURES FOR RQ1

In Table 8 we present the results for Precision, Recall and AUC.

**Table 8: Precision, Recall and AUC of LR models. The first section is Precision, the second section is Recall and the last one is AUC. TF refers to the dataset merged from Topi and FD, and TC refers to TREC + EvalCran.**

| | | Free-Form | | | Passage | | |
|---|---|---|---|---|---|---|---|
| | ALL | TF | Topi | FD | TC | TREC | Cran |
| $l$ | 0.668 | 0.665 | 0.665 | 0.000 | 0.875 | 1.000 | 0.833 |
| $s$ | 0.738 | 0.746 | 0.744 | 1.000 | 0.647 | 0.500 | 0.667 |
| $s_c$ | 0.690 | 0.696 | 0.688 | 1.000 | 0.667 | 0.750 | 0.647 |
| $n_{wtr}$ | 0.765 | 0.778 | 0.694 | 0.904 | 0.485 | 0.500 | 0.481 |
| $n_{ntr}$ | 0.679 | 0.690 | 0.674 | 0.884 | 0.485 | 0.500 | 0.481 |
| $n_{str}$ | 0.722 | 0.730 | 0.671 | 0.770 | 0.485 | 0.500 | 0.481 |
| $n_{wts}$ | 0.796 | 0.797 | 0.742 | 0.859 | 0.667 | 0.667 | 0.000 |
| $n_{nts}$ | 0.729 | 0.730 | 0.730 | 0.730 | 0.684 | 0.750 | 0.667 |
| $n_{sts}$ | 0.793 | 0.792 | 0.739 | 0.899 | 1.000 | 1.000 | 0.000 |
| $n_{wlt}$ | 0.475 | 0.475 | 0.502 | 0.467 | 0.485 | 0.500 | 0.481 |
| $n_{nlt}$ | 0.498 | 0.498 | 0.496 | 0.500 | 0.500 | 0.600 | 0.481 |
| $n_{slt}$ | 0.510 | 0.510 | 0.493 | 0.527 | 0.000 | 0.000 | 0.000 |
| $i$ | - | - | - | - | 0.679 | 0.833 | 0.786 |
| $r$ | - | - | - | - | 0.769 | 0.556 | 0.788 |
| $ii$ | - | - | - | - | 0.588 | 0.750 | 1.000 |
| $P(rel)$ | - | - | - | - | 0.571 | 0.500 | 0.542 |
| $l$ | 0.386 | 0.385 | 0.970 | 0.000 | 0.438 | 0.667 | 0.385 |
| $s$ | 0.166 | 0.157 | 0.391 | 0.004 | 0.688 | 0.333 | 0.769 |
| $s_c$ | 0.072 | 0.059 | 0.143 | 0.004 | 0.875 | 1.000 | 0.846 |
| $n_{wtr}$ | 0.613 | 0.607 | 0.819 | 0.467 | 1.000 | 1.000 | 1.000 |
| $n_{ntr}$ | 0.415 | 0.405 | 0.919 | 0.067 | 1.000 | 1.000 | 1.000 |
| $n_{str}$ | 0.776 | 0.772 | 0.714 | 0.811 | 1.000 | 1.000 | 1.000 |
| $n_{wts}$ | 0.714 | 0.724 | 0.908 | 0.604 | 0.125 | 0.667 | 0.000 |
| $n_{nts}$ | 0.693 | 0.691 | 0.577 | 0.766 | 0.812 | 1.000 | 0.769 |
| $n_{sts}$ | 0.597 | 0.604 | 0.946 | 0.379 | 0.188 | 1.000 | 0.000 |
| $n_{wlt}$ | 0.481 | 0.472 | 0.275 | 0.602 | 1.000 | 1.000 | 1.000 |
| $n_{nlt}$ | 0.325 | 0.313 | 0.332 | 0.301 | 1.000 | 1.000 | 1.000 |
| $n_{slt}$ | 0.743 | 0.755 | 0.933 | 0.639 | 0.000 | 0.000 | 0.000 |
| $i$ | - | - | - | - | 0.404 | 0.625 | 0.282 |
| $r$ | - | - | - | - | 0.638 | 0.625 | 0.667 |
| $ii$ | - | - | - | - | 0.213 | 0.375 | 0.077 |
| $P(rel)$ | - | - | - | - | 0.426 | 0.500 | 0.333 |
| $l$ | 0.773 | 0.773 | 0.865 | 0.975 | 0.795 | 1.000 | 0.778 |
| $s$ | 0.653 | 0.659 | 0.712 | 0.751 | 0.683 | 0.583 | 0.704 |
| $s_c$ | 0.613 | 0.613 | 0.630 | 0.751 | 0.801 | 0.875 | 0.784 |
| $n_{wtr}$ | 0.752 | 0.782 | 0.774 | 0.904 | 0.790 | 1.000 | 0.754 |
| $n_{ntr}$ | 0.696 | 0.718 | 0.775 | 0.817 | 0.794 | 1.000 | 0.749 |
| $n_{str}$ | 0.749 | 0.773 | 0.744 | 0.841 | 0.779 | 1.000 | 0.768 |
| $n_{wts}$ | 0.816 | 0.819 | 0.821 | 0.855 | 0.745 | 0.903 | 0.784 |
| $n_{nts}$ | 0.759 | 0.759 | 0.724 | 0.781 | 0.757 | 1.000 | 0.714 |
| $n_{sts}$ | 0.807 | 0.811 | 0.868 | 0.846 | 0.588 | 1.000 | 0.741 |
| $n_{wlt}$ | 0.485 | 0.484 | 0.502 | 0.468 | 0.467 | 0.378 | 0.452 |
| $n_{nlt}$ | 0.501 | 0.504 | 0.506 | 0.506 | 0.432 | 0.378 | 0.458 |
| $n_{slt}$ | 0.691 | 0.694 | 0.730 | 0.674 | 0.441 | 0.500 | 0.449 |
| $i$ | - | - | - | - | 0.628 | 0.835 | 0.710 |
| $r$ | - | - | - | - | 0.775 | 0.656 | 0.811 |
| $ii$ | - | - | - | - | 0.572 | 0.755 | 0.766 |
| $P(rel)$ | - | - | - | - | 0.562 | 0.561 | 0.523 |

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
