# OpenReview forum: "An Analysis of Stopping Strategies in Conversational Search Systems"
_ACM.org/SIGIR/ICTIR/2024/Conference — ICTIR 2024_

### Official Review · Reviewer_3LJK · 2024-05-16

**Rating:** 0
**Confidence:** 4

**Objective Part Of Review:**

This paper lays out and tests a few models of _stopping_ in conversational search: i.e., models that say when (and perhaps why) a searcher will disengage from a conversational system.
* Soundness: the overall idea (use established features to build logistic models and predict stopping in a bunch of corpora) is simple enough and seems sound. In particular, it is nice to see that the features chosen here are drawn from past work and from theory, not just from a fishing expedition, although the list in s2.3 could include features related to _rate of gain_ as seen in berry-picking and information-foraging models (see e.g. Pirolli and Card, Psych Rev 106(4), 1999; or Azzopardi et al., SIGIR'18, for an explicit stopping rule).
The list of rules in s4 however doesn't line up with the list of rules in s3, or if they do then I missed how. SR1 doesn't appear in any of the rules in s3; SR2 is perhaps relevant to difference threshold, as is SR3, but it's not clear which is more faithful to Nickles's idea; and so on. Explicit linking text might help here, and/or even a little diagram might (I drew one myself as I was reading). It does seem that some rules from s3 -- mental list, magnitude threshold, representational stability, and maybe the combination rule -- aren't covered in these features, and similar some features (SR5, SR7, SR8-10) don't draw on any of the theories discussed. SR8-10 in particular feel ad-hoc: some justification here, presumably linking the noun features to some of the rules via Clark and Sengul, would help.
The data sets are useful, but vary a lot (long vs short text, retrieval vs generation, ...). This might explain why the final models look so different. But it's also critical to know more about how each was collected, since none are naturally-occuring conversations. The instructions given to conversants, and the incentives to go longer or stop early, seem absolutely critical here. Interaction details such as response speed and mode would also make a big difference. More details in the paper would perhaps help us guess what features would be predictive of stopping behaviour, and help to validate the final models: as it is, it's hard to know whether the final models seem valid or not.
The fitted models could also be subject to e.g. backward model selection or ablation, to simplify them and focus more on what's important. Similarly, reporting e.g. partial eta-squared (or some similar statistic) would help understand the importance of different features -- the manipulations in s6.3 could perhaps be removed in favour of something like this. (Related, why care about early importance if we're on our way to building a better model anyway? This isn't clear.)
* Presentation: this is an easy read (with some minor points noted below). The purpose of s6.3 wasn't entirely clear, and there are a few small typos.
* Difficulty: straightforward work, but that does not make it any less important.
* Related work: a brief but useful survey is included.

**Subjective Part Of Review:**

This is an interesting topic, and it's nice to see tests of the ideas with conversational corpora.
One problem with this flavour of work in general is that we're measuring current systems -- possibly even out of date, already -- instead of whatever's next. It would be nice to read some thoughts on whether we can use (or build) a theory to generalise to tomorrow's systems. For example, it might be that disfluencies are a problem, so a longer conversation means people are more likely to give up. If we fix the disfluencies, that correlation would go away. Are the criteria listed here generalisable? Are the models? (Certainly they seem very different from one corpus to the next.)
Some smaller thoughts:
* abstract: does conversation really have a "unique sequential and interactive nature"? So does a series of e.g. conventional web searches, with clicks and other interactions.
* s1, para 1: "with the exception of ad-hoc retrieval": I think you mean to say the opposite, ad-hoc retrieval is more or less defined by handling queries independently.
* s1 and throughout: is "query" the best term in a CSS? Perhaps "utterance"?
* s1, para 2: "One of the main challenges...": this is a challenge for the searcher, not the system, unless the system aborts early.
 * s3, RQ3: "contribute to" should probably be "correlate with" unless we're making an explicitly causal claim.
* s6.2, para 5: "satisfactory F1 scores": how is "satisfactory" determined?
* s6 throughout: different names are used for the same data: e.g. "TF" amd "Topi+FD" in Tables 4 and 5.
* Table 5 is an interesting idea.

---

### Official Review · Reviewer_s1Zh · 2024-05-16

**Rating:** 2
**Confidence:** 4

**Objective Part Of Review:**

Overview: This paper investigates stopping strategies in conversational search systems (CSSs), adapting traditional IR stopping rules and proposing new ones based on textual features. Experiments on multiple datasets demonstrate the effectiveness of these rules in predicting user stopping points.

Soundness: The paper is methodologically sound, providing clear explanations of the adapted and proposed stopping rules, experimental setup, and results. The claims are supported by empirical evidence and logical reasoning.

Difficulty: The paper addresses a complex problem in a methodologically sound way. The research is significant as it explores a novel area in IR, contributing valuable insights into user behavior in conversational systems.

Methodology: The paper presents a rigorous methodology, adapting existing stopping rules and proposing new ones. The approach is sound and comprehensive, combining theoretical and modeling aspects with empirical validation using multiple datasets. The methodology employed is robust, involving the adaptation of existing SRs to CSSs, the creation of new SRs, and the identification of key features predictive of stopping points in conversations. This approach is not merely an add-on but forms the core theoretical contribution of the paper, integrating conceptual and modeling work to address the unique challenges posed by CSSs.

Related Work: The paper provides a thorough review of relevant literature. The related work section of the paper provides a comprehensive overview of the existing literature in conversational search systems (CSS) and stopping strategies within information retrieval (IR). The authors have effectively highlighted the evolution of conversational search, user modeling, and the development of stopping rules, which are crucial for understanding the context of their research. They have cited a range of seminal and recent works, which helps in establishing the relevance and timeliness of their study.

Conclusion: The paper concludes that understanding and modeling stopping behavior in conversational search systems is essential for both theoretical advancements and practical applications in information retrieval. The adapted and newly proposed stopping rules not only enhance the understanding of user interaction patterns but also offer a methodological framework for future research to build upon. This study marks a significant step towards creating more engaging and efficient conversational search systems by aligning system responses more closely with user expectations and behaviors.

Key Claims and Observations
1. **Adaptation of Stopping Rules**: The paper successfully adapts traditional IR stopping rules to the conversational context, acknowledging the unique challenges posed by the interactive and sequential nature of CSSs.
2. **Development of New Stopping Rules**: New stopping rules are proposed based on textual statistical features such as the number of words, nouns, noun phrases, and sentences in user-system interactions. These features significantly influence the determination of stopping points.
3. **Importance of Textual Statistical Features**: The study finds that textual statistical features play a critical role in predicting when users will stop their interactions with CSSs. Specifically, the number of unique nouns outputted by users is highlighted as a particularly influential feature.

Findings and Takeaways
1. **Performance of Stopping Rules**: The adapted and newly developed stopping rules show promising results in predicting stopping points across several conversational datasets. This indicates the potential of these rules to enhance the realism of user simulations.
2. **Foundation for Future Research**: The results provide a foundational understanding that can guide the design of more reliable evaluation measures for CSSs and contribute to the development of more sophisticated user models and simulators.
3. **Guidance for CSS Design**: Insights from the analysis can help in designing CSSs that are better aligned with user behaviors and preferences, potentially improving user satisfaction and system effectiveness. Overview: This paper investigates stopping strategies in conversational search systems (CSSs), adapting traditional IR stopping rules and proposing new ones based on textual features. Experiments on multiple datasets demonstrate the effectiveness of these rules in predicting user stopping points.


## quick answers to objective review questions

•	Problem Statement: the problem dealing with the lack of effective stopping strategies and predicting stopping points in conversational search systems (CSSs) is clearly articulated.
•	Methods: the paper thoroughly describes the adaptation of IR stopping rules to CSSs, datasets used, features extracted, and logistic regression modeling approach.
•	Results: the findings on stopping rule performance, feature influence, and model evaluation metrics are clearly presented.
•	Claims: the claims are supported by empirical evidence and discussions based on experiments with multiple datasets.
•	Definitions/Abbreviations: all concepts and notations are defined before use, ensuring clarity and understanding.
•	Abstract/Introduction: Yes, both provide a clear overview of the study's goals, significance, methods, and findings.
•	Contradictions/Errors: No apparent contradictions or errors were found.
•	Relevant Work Cited:	the paper cites relevant and competing works comprehensively, contextualizing the research within the current literature.
Overall: The paper is well-structured and contributes new insights into stopping strategies in conversational search systems, effectively situating itself within existing literature.


## Immediate areas for improvement:

- Page 1:
    - “the recency effect, which significantly influences user satisfaction. “
        - Can you provide an example or cite a reference here please?
- Page 4:
    - “We merged TopiOCQA, and FAITHDIAL since both of them feature free-form as the response style, as well as TREC CAsT 2021 and EvalCran feature raw passages as the response. “
        - The sentence may be split for better readability
- Page 6:
    - It might help novice researchers to understand why min-max normalization was used (as against standardization) for featurization before building the LR model
- Page 7:
    - Figure 2 and Figure 3 legend fonts need to be bigger please.

## Immediate areas of concern
None

## Suggested areas for improvement
### 4 STOPPING RULES
Given the outlined method, one could formulate an arbitrary number of stopping rules with increasing levels of complexity (e.g. going from nouns to entities) or relatedness (connections between entities). Depending on the conversation, one could also account for the effect of sentiment/emotional state/time constraints that might influence a user’s stopping rule.
Understandably the results of this study are more widely applicable in a generic sense, however, it would be interesting to factor in the users’ mindset, as well as, investigate increased complexity of stopping rules. I wonder if we could build a model to predict a stopping rule given appropriate user context, type of information need, and the available information?

Although not directly related to conversational search systems, the Technology Assisted Review (TAR) methods for enterprise search and e-discovery especially for high-recall information retrieval, have dealt with the analysis of __stopping criteria__ [G. V. Cormack and M. R. Grossman. Engineering quality and reliability in technology-assisted review. In SIGIR 2016. ; (and other similar papers)]
The “conversation” in the TAR sense is between a human and a continuous active learning classifier that is being trained to return relevant documents given labels provided by a human expert in the loop. The human assessor reviews a sample of predicted relevant documents, and returns rel/non-rel labels back to the classifier for retraining before the next iteration.


### 5.2 Grouping, Splitting and Augmenting
Incomplete conversation need not be the same as unsuccessful stop. Does your dataset have
true negative samples where the user stopped without taking in all the relevant info or stopped
before getting frustrated? Maybe the true negatives don’t matter because the users may
abandon the conversation [Abdigani Diriye, Ryen W. White, Georg Buscher, S. Dumais
“Leaving so soon?: understanding and predicting web search abandonment rationales”, CIKM 2012.; (and other similar studies on abandonment)]

Of course, the analysis of abandonment in conversational search systems is still nascent area. However, it would be good to have a paragraph discussing the true negative scenarios in comparison to the incomplete conversations in the augmented dataset.

### 6 EXPERIMENTS AND FINDINGS
It would be interesting to look at the interplay between nouns in relevant and nouns in non-relevant responses; could there be a different in stopping rule feature importances?

**Subjective Part Of Review:**

- **Readability and Understandability**: The paper is well-structured and detailed in its approach to adapting stopping rules from traditional information retrieval systems to conversational search systems. The authors provide a clear introduction to the problem, a comprehensive review of related work, and a detailed explanation of their methodology and experiments. Overall, the paper is well-written!

- **Relevance of the Problem**: The problem addressed by the paper is highly relevant. As conversational systems become increasingly prevalent, understanding when users choose to stop interacting with these systems is crucial for improving user experience and system efficiency. The focus on stopping strategies in conversational search systems fills a significant gap in the existing research, making it a timely and important study.

- **Originality of Methods**: The paper's approach to adapting and analyzing stopping rules in the context of conversational search systems is original. The adaptation of traditional stopping rules to a conversational setting and the introduction of new rules based on conversational features demonstrate innovative thinking. Moreover, the use of multiple datasets to validate the effectiveness of these rules adds depth to the research.

- **Interest of Results**: The results are interesting and provide valuable insights into the factors that influence stopping behavior in conversational search systems. The identification of key textual statistical features and the impact of unique nouns on stopping points are particularly noteworthy findings that contribute to our understanding of user behavior in conversational contexts.

- **Interest to the ICTIR Community**: This work is likely to be of significant interest to the ICTIR community. It addresses an important aspect of conversational search systems, an area of growing importance in information retrieval. The findings on stopping strategies have implications for the design of more effective conversational agents and can influence future research on user interaction with AI-driven systems. The methodological innovations and comprehensive analysis make this paper a valuable contribution to the field.

---

### Official Review · Reviewer_8ynJ · 2024-05-17

**Rating:** -1
**Confidence:** 4

**Objective Part Of Review:**

This submission to ICTIR 2024 analyzes different stopping strategies in conversational search simulations. The authors evaluate different features/combinations of features on four different test collections to learn about their influence on the correct stopping prediction.

Some open questions I had, after reading the paper:
- TF-IDF is used to compare document similarity? How? By using the dot product? How is the threshold of 0.5 chosen?
- None of the results seems to be tested on significance, although in the text formulations like "... only three features change significantly..." appear. Please clarify and check the actual significance.
- Why is the performance of l, s, and s_c in FD (table 3) near 0 but the highest influential component (in the case of l) in Table 6? How does this match?
- Presentation is an issue; many abbreviations for the different features are introduced, but it is hard for the reader to keep along. I flipped pages back and forth as I forgot about the difference between n_{ntr} and n_{str}. This might also be related to the high number of features and the overall exhaustive description. Maybe a more condensed version of the paper would allow to boil it down slightly more.

**Subjective Part Of Review:**

Overall, this is a nice paper that does an insightful evaluation of stopping criteria that can be useful for later work on user simulations in conversational systems. The writing is easy to follow (although it might be a little bit more condensed). The tables and figures are mostly easy to follow as well.

I do miss the theoretical aspects in this work. The paper reads like a typical analytical/experimentation paper that evaluates some features and their performance on different data sets. While this is absolutely valid and insightful to a specific degree, I would love to see a theoretical discussion on some new aspects, such as human factors, cognitive aspects, etc., of stopping criteria that would add something new to the discussion. For example, it was mentioned that relevance information can not be exchanged with other features. That sounds like a very interesting outcome, but it is not discussed (and many other aspects are not discussed either) in the paper.

I miss the depth, meta-insights, or new ideas/thoughts a good ICTIR should contain. I am therefore against accepting the paper for ICITIR. Its core part is solid but limited in the insights provided, and the theoretical dimension is missing at all.

---

### Official Review · Reviewer_bTLH · 2024-05-18

**Rating:** 1
**Confidence:** 5

**Objective Part Of Review:**

1	Objective comments, general

This paper describes a study aimed at developing rules for identifying when a conversational search will stop. The primary goal of this research, well stated and motivated, is to provide information necessary for accurate and realistic simulation of conversational search. There is also the suggestion that the results could influence the design of conversational search systems. The study itself is well designed and described, and the results, although not terribly surprising, are of some interest, and identify rules that go well beyond those which have previously been developed for non-conversational search.

There are a few issues arising in the presentation that are somewhat problematic, but which can easily be addressed. These are identified in the specific comments, below.

1.1	Objective comments, specific

Section 1, paragraph 1, “with the exception of ad hoc retrieval”: In what sense is "ad hoc retrieval" any different from the typical query submitted to a search system? Ad hoc retrieval is the term that’s been used consistently to describe standard search, so it’s quite unclear what the authors mean to convey here.

Section 1, paragraph 1, “queries form sequences like conversations, making this independence assumption unrealistic.”: Why is it any more unrealistic than in ordinary search systems, in which a large body of research demonstrates that search sessions, containing a number of related queries, are a frequent occurrence.

Section 2.1, paragraph 1, last sentence: Neither of the cited papers actually deals with the use of information by the searcher, which I take this sentence to refer to. So what is meant by "usage of information" here?

Section 2.3: I’m surprised not to see this paper cited here:
Ilani, F., Nowkarizi, M., & Arastoopoor, S. (2023). Analysis of the factors affecting information search stopping behavior: A systematic review. Journal of Librarianship and Information Science, 0(0). https://doi.org/10.1177/09610006231157091

Section 5, Table 1, Cell EvalCran/Answer Style: I think that this should be "Paragraph", rather than Passage, since the answer is not an extract from a text, but a complete paragraph. This means that this dataset contains substantially more nouns/noun phrases per reply and conversation, than the others, which is an important consideration for the SRs.

Section 6.3, paragraph 3, sentence 2: The three should be identified here, not just referred to.

Section 6.3, paragraph 3, sentence 3: It's very difficult for me to understand how features associated with the last turn in a conversation can possibly be indicative of a continuation of the conversation. I think this needs more detailed comment than is given here.

Section 6.3, paragraph 3, sentence 4: In what table or figure is this noticable? And is "increase of data" an increase in percentage of data used, or in absolute size of database?

Section 7, Results, point (5): “necessarily contribution” doesn’t make sense; what is this SR?

**Subjective Part Of Review:**

I think that the results provided here will be of some interest to ICTIR attendees, and to the conversations search system community at large, especially with respect to developing more accurate and realistic simulations of conversational search. This could also be important for developing measures for design and evaluation of conversational search systems.

---

### Meta-Review · Area_Chair_hhhL · 2024-06-02

**Recommendation:** Accept (Oral)
**Confidence:** 5

**Metareview:**

The work effectively utilizes the features proposed in other theory-driven works in a slightly different setting. The reviewers appreciate the theory in extending and modifying stopping behavior, especially with respect to the statistical and linguistic features of the conversation.

There are several suggestions on how the paper can be improved. In particular, I would like to point to the potential of further discussion of some features and the generalizability of the work. Moreover, some discussions can be added on the implications of these stopping criteria on the future generations of conversational search systems would be appreciated. As mentioned by 3LJK, some of these systems have already expired and some of the assumptions on older models do not necessarily hold on the next generations. Therefore, the authors could improve their paper by having a discussion on these aspects.